# Carbon removal support is tempered by concerns over whether biological methods are worth it
Emily Cox [1,2] ✉, Laurie Waller [3], James Palmer [4] & Rob Bellamy [3] ✉

Biological carbon removal has been proposed as a 'win-win' for climate, sustainability and public opinion, but research on public perceptions is lacking explicit evidence on trade-offs between options. Here we explore perceptions using small group deliberation (n60) plus a nationally representative survey (n2027) in the UK's four jurisdictions. We find a strong preference for carbon removal to play a substantial role in meeting national climate targets, stemming from persistent scepticism about emissions reductions and behaviour change. However, such support was tempered with caution about whether certain biological techniques - biochar, peatland restoration, and perennial biomass crops - would be "worth it". In particular, concerns were raised about life-cycle emissions, as well as land competition with urgent housing needs, and scientific uncertainty around novel techniques such as biochar. While we find that responses to carbon removal tend to shift the burden of responsibility for climate action away from individuals, we also identify region-specific discourses, highlighting the importance of local context in shaping public views.

In the race to scale up carbon removal to Gigatonne scale by mid-century[1], so-called 'nature-based' carbon removal solutions occupy a large and growing role in policy debates and the public imagination[2,3]. Capturing and storing $CO_2$ via biological processes – in vegetation, soils and sediments – has been proposed to provide 'win-win' solutions for climate and sustainability, whilst potentially sequestering considerable proportions of anthropogenic carbon emissions[4,5]. However, such accounts threaten to obscure concerns about likely risks and trade-offs[3], occasionally over-emphasising promissory claims and straying into exaggeration and hype[6,7].

Terms such as 'nature-based' create strong framing effects[8,9], and flatten important distinctions between the method, removal process and storage medium, therefore we prefer to use the term 'biological' carbon dioxide removal (CDR). Biological CDR techniques are the most prevalent in the scientific literature[10], but have received less attention from social scientists, particularly lacking data on public perceptions of techniques such as biochar, perennial biomass crops and peatland restoration[11]. We know that techniques perceived as more 'natural' are likely to be preferred, but this has mainly been demonstrated with respect to so-called 'conventional' CDR techniques such as afforestation and soil carbon sequestration[2,12], (cf. ref. 10 for the conventional vs novel CDR distinction). A few studies have examined perceptions of biochar, finding that it is generally preferred over non-biological CDR methods, but less preferred to conventional CDR[13–15]. This

literature, however, has generally focused on farmers' perspectives, largely ignoring broader societal viewpoints[6]. Meanwhile a larger body of literature examines public perceptions of Bioenergy with Carbon Capture and Storage (BECCS), finding substantial concerns about geological storage of $CO_2$, land use and competition with food production[2,16–18]. By contrast, this research explores public perceptions of a suite of both 'novel' and 'conventional' biological methods which have been less well studied—biochar, perennial biomass crops, and peatland restoration—and which raise distinct technical and social challenges (cf. ref. 10). Since none of these are currently deployed for large-scale carbon removal in the UK, we expected them to encounter similarly low levels of familiarity.

Previous research on public perceptions of Carbon Dioxide Removal (CDR) argues that CDR should be viewed 'in context' as just one of many possible interventions for tackling climate change. However, a surprising lack of work explicitly interrogates the potential trade-offs that this might entail. All policy decisions have opportunity costs, and there is ongoing debate over how much of a role CDR should play compared to emissions reductions. Which emissions are deemed 'hard to abate' is an inherently political and social question[19,20], suggesting there is an important public conversation to be had. Therefore this research aims to examine public views on the preferred balance of CDR and emissions reductions in meeting national climate targets.

[1]Smith School of Enterprise and the Environment, University of Oxford, Oxford, UK. [2]Understanding Risk Group, School of Psychology, Cardiff University, Cardiff, Wales, UK. [3]Department of Geography, University of Manchester, Manchester, UK. [4]School of Geographical Sciences, University of Bristol, Bristol, UK. ✉e-mail: emily.cox@cse.org.uk; rob.bellamy@manchester.ac.uk

https://doi.org/10.1038/s43247-025-02654-x                                                                                                    **Article**

From the above, we develop three overarching goals of this research:

1. To understand public attitudes and discourses on novel and conventional biological CDR, with a focus on biochar, perennial biomass crops and peatland restoration
2. To interrogate potential trade-offs between CDR and emissions reductions, and elicit public views on the preferred balance
3. To explore region-specific discourses about CDR, and compare how responsibility for climate action is perceived in different jurisdictions of the UK.

In-line with well-established practices for researching perceptions of novel technologies, we use in-depth deliberation amongst small groups. However, a recognised problem concerns how researchers should 'frame' the topic in order to facilitate discussions, which influences the outcomes[21]. As an 'upstream' domain of science and technology subject to considerable uncertainties and ambiguities, CDR methods are undoubtedly susceptible to framing effects. One key implication of this is that deliberative research should seek to 'open up' and 'unframe' discussions[22]. We tackle this challenge by combining physical samples and internet-generated images alongside textual descriptions of methods, enabling participants to engage with the carbon removal methods and articulate relevant experiences in multiple ways, beyond cognitive responses. We also organise our workshops around two distinct framings – one 'techno-economic' frame which centres quantitative CDR scenarios[23,24] (Group 1), and one 'everyday life' frame emphasising the importance of personal experience to environmental knowledge (Group 2), enabling comparison of discursive frames across different locations. We held four deliberative workshops ($n = 60$) in the four devolved jurisdictions of the UK (Scotland, Northern Ireland, Wales and England), each split into the two framing groups for the duration. The choice of location reflects the devolved nature of land policy in the UK. In parallel, we also conducted a nationally-representative survey of the UK public ($n = 2027$) with samples weighted according to population size in each jurisdiction. The survey tested knowledge and perceptions of the three techniques, and CDR in general, with perennial biomass crops split into soil carbon sequestration (SCS) and BECCS; deliberative materials also presented these two options for the PBC $CO_2$ storage (Supplemental 5).

## Results

In the survey, a large majority of participants were worried about climate change, and 50.3% expressed serious worry (8 or above on a 1–10 scale). 77.8% said they were aware of the UK's net zero target, and of these, 78.7% supported it, with only 8.3% expressing opposition to the net zero target (Supplemental 7). This mirrored the deliberative workshops, where we opened with a discussion on net zero (see Methods), and found that all groups were concerned about climate change and highly supportive of doing more to tackle it, although they became more ambivalent when considering specific responses and policies.

Survey respondents reported low prior awareness of the CDR techniques. Self-reported knowledge (Fig. 1) was highest for CDR in general and for peatland restoration, with 31.4% and 32.4% of survey participants respectively saying they know at least a moderate amount, and lowest for biochar, with 75.3% of the survey saying they 'know nothing' or 'have never heard of' biochar before. A one-way repeated measures ANOVA (two-tailed) showed that these differences were statistically significant, $F_{(3.72, 7526.34)} = 83.85$, $p \le 0.001$, partial $\eta^2 = 0.252$, with all pairwise comparisons (Bonferroni) statistically significant at $p \le 0.001$.

### Exploring attitudes and discourses to biological CDR techniques

In the deliberative workshops, there were many similarities between the three techniques, with the major themes cutting across all three. The strongest theme which emerged from the thematic analysis – in all 8 workshop groups – was the question, "Is it worth it?", and it is this theme which we focus on in this section. For instance, Isla (Belfast Group 2) said: "*I*

*would just want to know is it worth it. Just, like, economy wise, like, $CO_2$ wise, is it worth it?*". This concern had many layers, and participants evaluated the economic 'worth' of carbon removal in relation to diverse arenas of social life and engagements with the environment, highlighting notions of the value of CDR that may not be easily quantified or modelled. Indeed, whether carbon removal was perceived to be worthwhile was often considered separable from questions of its cost.

That said, a more unexpected finding was the way that all groups most frequently focused on 'worth' in terms of whether more $CO_2$ was being absorbed and stored during the entire life-cycle of the process: "*And the net result is how much of a reduction would it be, because forget the cost, how much of a reduction would you get?*" (Seamus, Cardiff G2). This focus occurred equally in the 'everyday life' groups, where we consciously sought to avoid such techno-economic framings. Participants also raised concerns about the durability of the carbon stored, both in terms of natural instability: "*We could have a drought one minute – look at the weather this September… and any more carbon then that's going into the atmosphere because you can't maintain it well*" (Rowan, London G1), and socio-political instability: "*You're entrusting the successive governments and generations will honour that system, and it doesn't really seem like that's going to happen, because the moment land's needed for something, it will be…and then all the carbon's released [Laughs]*". (Gabriel, Belfast G1).

Questions of 'worth' were also rooted in concerns about cost and trade-offs against other policy spending objectives: "*In the current climate though, there's no money to spare… not just spend all this money that you're going to take from somewhere that desperately needs it*" (Ellie, Belfast G1). In common with other studies, we identified concerns about land use and food production; yet participants also focused on housing shortages, including directly prioritising a perceived housing emergency over the climate: "*Land is at a shortage… the population is exploding and people are homeless and have nowhere to live. Why not build housing or temporary housing on that land? Are there better uses for that land which are more needed for other emergencies?*" (Lily, London G2). Participants suggested that these CDR techniques would encounter public opposition, whilst deliberately distancing themselves from "*You know, the naysayers, people who say, oh, it won't work because…*" (David, Belfast G1).

Our thematic analysis identified a number of sub-themes sitting within the theme of 'is it worth it'. The most prevalent of these, across all groups, was around scientific uncertainty. This was particularly the case for biochar, with similar discourses across both framing groups, and was instrumental in generating relatively more caution toward biochar than the other two techniques: "*I still don't really think there's enough research, like there's not really enough information…*" (Louise, Belfast G1). This was supported by the survey data, wherein biochar encountered slightly more negativity, but also much more uncertainty (Fig. 2). A one-way repeated measures ANOVA (two-tailed, Greenhouse-Geiser correction) showed that these differences were statistically significant, $F_{(3.73, 4675.78)} = 209.31$, $p \le 0.001$, partial $\eta^2 = 0.143$. (Those who answered that they "don't know enough" about the technique to answer this question were excluded listwise from the sample, leaving a sample size of $n = 1254$; all pairwise comparisons (Bonferroni) were statistically significant at $p \le 0.001$).

Uncertainty was portrayed as not just a scientific question but a moral one with implications for accountability and liability – in other words, who bears the responsibility and cost if a technique fails to work as planned at scale? As Dani and Ruby (Edinburgh G1) said about biochar: "*It almost makes me think that we can trust it as a solution, for the long term… And no one is being held accountable if it doesn't work the way we want it*". For all three techniques, participants expressed a desire for empirical data from real-world studies, yet also engaged in fascinating debates about whether professionalised expertise is less trustworthy than traditional or landholder knowledge: "*I've just got a feeling that I don't trust the experts. I think there's money involved in this, you know… And like you said, farmers have been doing it for so long… They know their field, don't they?*" (Raul, Olivia & Sarah, Cardiff G1).

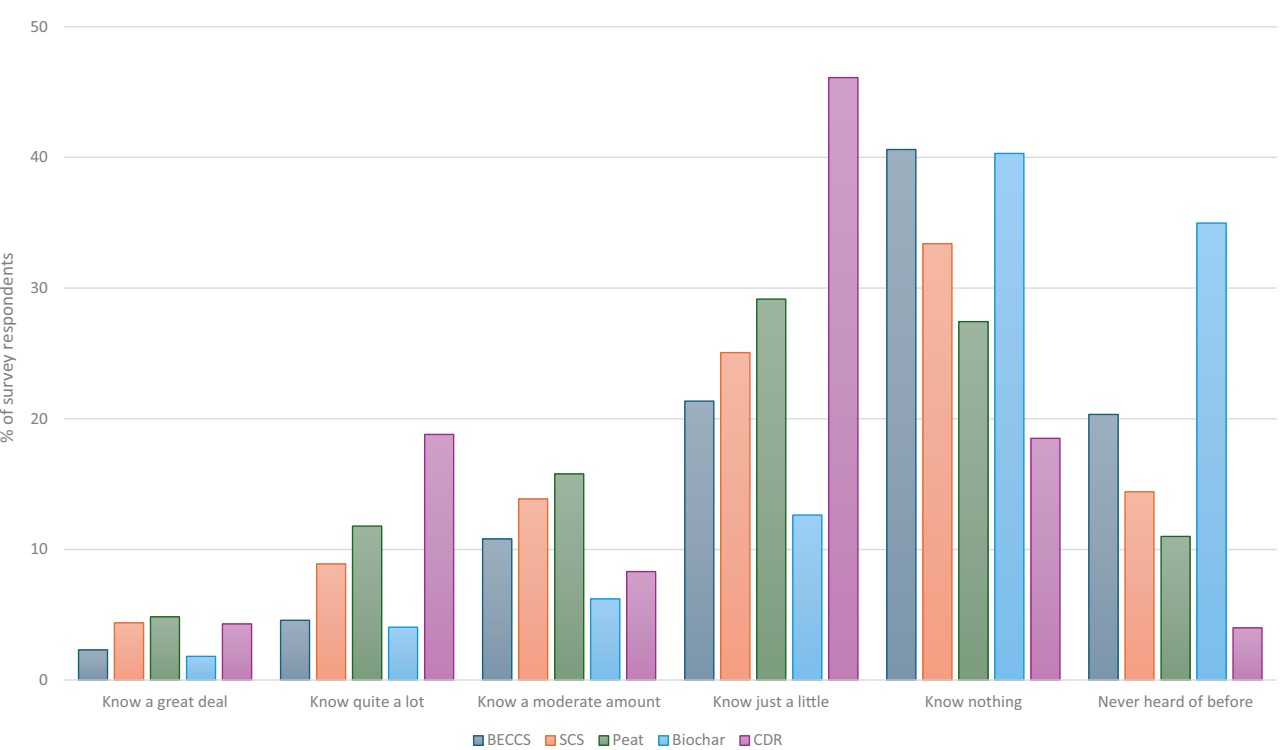

**Fig. 1 | Self-reported knowledge.** Survey responses to the question "Before today, how much if anything would you say that you know about carbon removal?" (*n* = 2027).

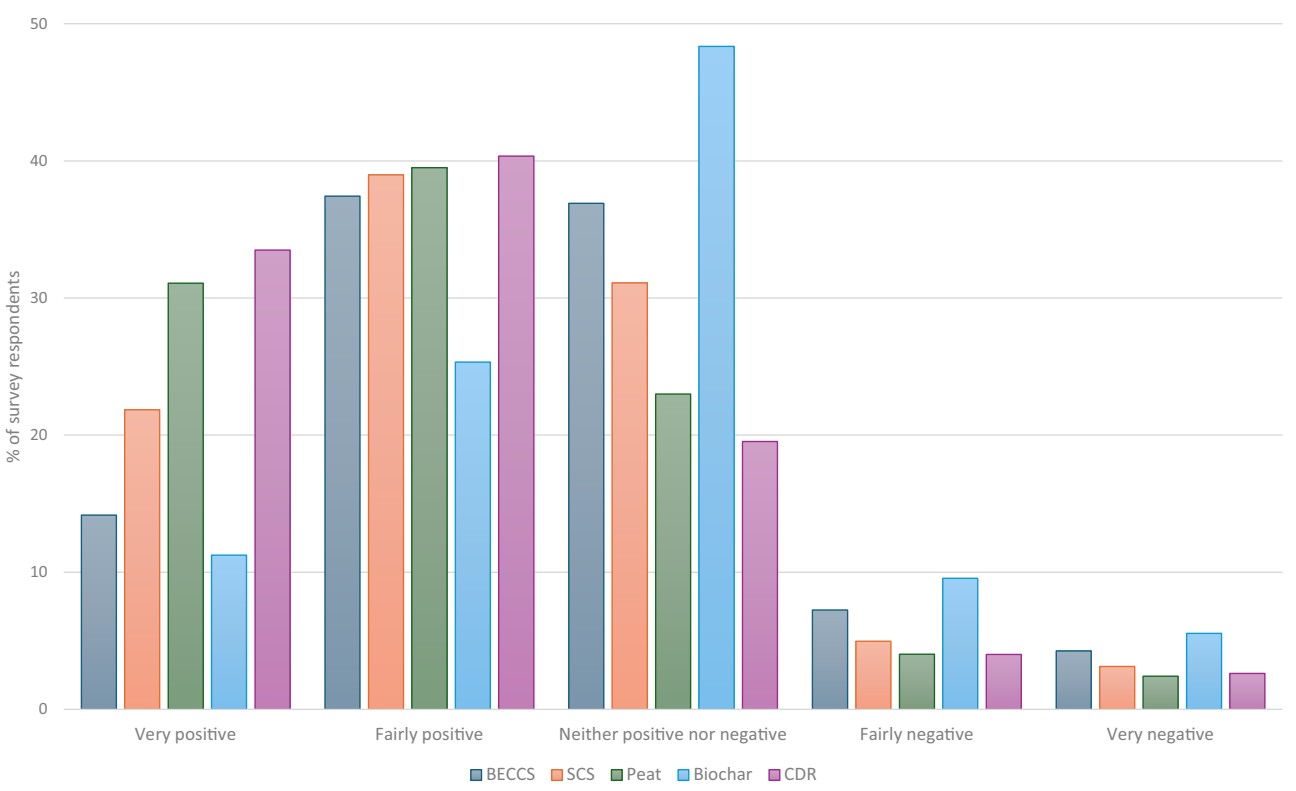

**Fig. 2 | Attitudes to biological CDR techniques and to CDR in general.** Survey responses to the question "How do you feel about [technique]?" Participants who answered "don't know enough about it" excluded.

At the end of the workshop, we ran an activity designed to make groups compare the three techniques, in a way which encouraged consideration of trade-offs and opportunity costs. Groups were given 9 sticky dots, and asked to discuss and to reach consensus on how many dots to assign each technique, according to the preferred *relative* role of each in meeting the jurisdiction's climate targets (Fig. 3). Consistent with survey results, biochar was least preferred overall. When forced to consider trade-offs between the techniques, PBC was most preferred by the majority of groups.

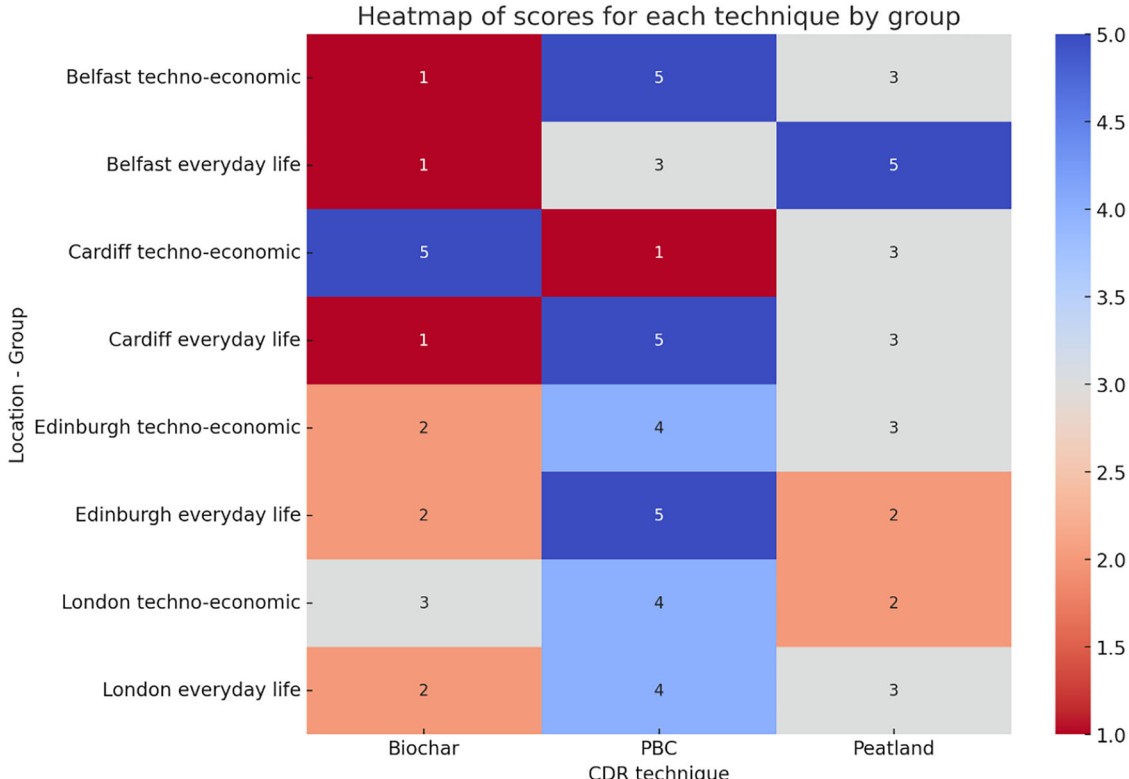

**Fig. 3 | Rank scores for the three techniques.** Heatmap showing the results of the ranking task. Groups asked to reach consensus on assigning 9 sticky dots to the 3 techniques (biochar, perennial biomass crops and peatland restoration), corresponding to how much of a role they should play compared to one another in meeting the jurisdiction's climate targets. All rows add up to 9.

## Trade-offs between CDR and emissions reductions: scenario activity

One of the key findings from our work was a strong preference for CDR to play a substantial role in meeting each jurisdiction's climate targets, although this was tempered with caution about whether the specific biological techniques would be "worth it". These findings are supported by the survey results (Fig. 2), which find that CDR in general is significantly preferred over the individual techniques, confirmed by a repeated-measures t-test (two-tailed), $t(1783) = -18.266$, $p \leq 0.001$ [−0.435, −0.351] (BCa bootstrapped to 1000 samples).

In the workshops, we sought to provoke discussion about the balance of reductions and removals, and the trade-offs involved, via an activity where participants were asked to choose between four fictional scenarios for the proportion of CDR in meeting national climate targets (see Methods). The goal here was to encourage discussion and critical engagement with the idea of trade-offs, rather than the numerical content of the scenarios, and we emphasised this to participants. Overall, we found that none of the workshop groups – and very few individual participants – felt that CDR should not be used.

The first time we ran the scenarios exercise, early on in the workshops, the majority opted for the scenario with the largest proportion of CDR (Fig. 4). The main reason given was the perceived difficulty of reducing emissions, which was seen as being entirely synonymous with individual behaviour change. Such individualisation of responsibility for tackling climate change meant that emission reduction targets were seen as too difficult and too societally problematic: *"90% [emissions reductions by 2050] sounds very high and is it achievable? I can't imagine putting a huge thing like that on people. I don't think you'd get a lot of buy-in, you'd get a lot of negativity."* (Ellie, Belfast G1). By contrast, the role of polluting industries and fossil extractivism in emissions reductions were barely mentioned, and did not play a major role in group discussions about how to rank the scenarios.

Some participants even framed CDR as an option which could be undertaken by businesses and companies, taking some of the pressure and responsibility off individuals: *"When you say, reduce emissions, it kind of feels more like, the community is more responsible, and then removal is more towards businesses"* (Dani, Edinburgh G2).

Several hours later, we ran this exercise a second time, after the detailed discussion of the three CDR techniques. Here, some of the initial high expectations about CDR were scaled back, particularly due to participants' concerns about the scale and durability of the emissions removals: *"I also changed to [a scenario with less CDR] because I thought the carbon removal to me is almost going to create more carbon in the first instance"* (Louise, Belfast G1). Three participants remained sceptical about CDR throughout the workshops: *"It's got to be easier to fix the cause, I mean, anything else is like sticking a very small sticking plaster over a huge gaping wound". (Tim, London G2).* Yet the majority continued to discuss CDR in the context of the difficulties of behaviour change, again demonstrating the dominance of individualised discourses on emissions reductions: *"It's just, I don't think you can fully rely on humans to change their natural instincts, and what they've been doing. So, there should be some reliance on carbon removal, that we don't really control as a society"* (Jack, Edinburgh G1). Thus the groups continued to support a large role for CDR, albeit with evident reluctant acceptance (cf. ref. 25): *"I wasn't convinced by any of these [CDR techniques] by a long shot, but I just feel like reducing emissions is such a hard task for everyday life for people, so I feel like going [more CDR] for now just seems to be the most pragmatic option (Joseph, London G2)".* Figure 4 shows the individual responses to the four fictional scenarios, demonstrating the way in which initial expectations of CDR were scaled back during the course of the workshop, yet continuing to reject the notion of relying on emissions reductions to meet 90% or 100% of emissions targets.

Overall, as Fig. 4 illustrates, most participants started the workshop with low hopes for mitigation and high expectations of CDR; as they learnt

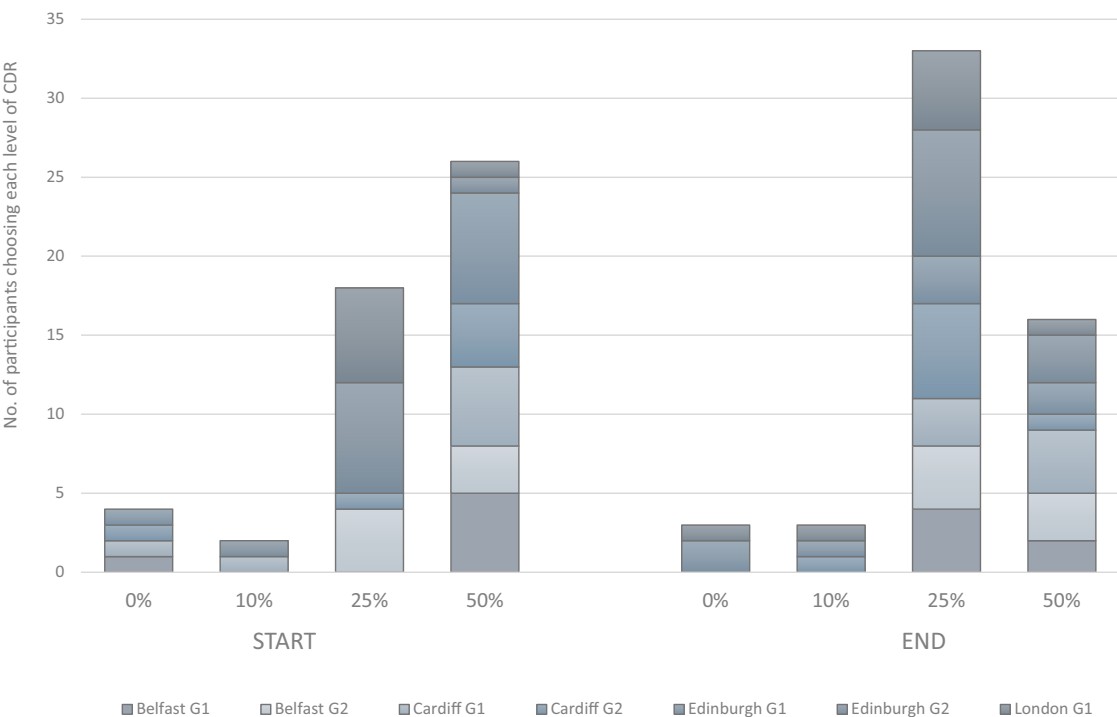

**Fig. 4 | Support for different levels of CDR at the start and end of the workshop.** Individual responses to the scenarios comparison, near the start of the workshop (following brief introduction to CDR and general discussion), and near the end (following in-depth discussions of the three biological techniques) (see Supplemental 4, 6). Start responses indicated by a discussion and a show of hands; End responses indicated by a discussion and a questionnaire. Participants given four fictional scenarios to choose from, with differing percentages of CDR versus emissions reductions in meeting net zero emissions targets (Methods). Horizontal axis shows the % of CDR in each scenario. London Group 2 data missing.

about CDR techniques, many became slightly more sceptical about CDR, yet the scepticism about emissions reductions remained stronger. This led to proposals for a broad portfolio of options: *"I just don't think there's a magic bullet that's going to solve… it's going to take a lot of small and big things to get us where we need to be"* (Gabriel, Belfast G1). In particular, participants emphasised the importance of considering different timescales. Multiple groups proposed a phased approach, with CDR being used in the short term not just to buy time, but also — in an interesting twist on conventional theories of "mitigation deterrence"[26] — to help build societal support for the 'more challenging' emissions reductions in the longer term: *"Because you're partially focusing on removal, the people that would be resistant to change or who would take a little bit longer to adapt, it's almost like offsetting for the people who might be a little bit behind the curve"* (Lucy, Cardiff G1). Emission reduction was seen as an "aspiration" (Mark, London G2), but not necessarily possible in the near-term, demonstrating the way in which participants made trade-offs between the desirable and the feasible, before eventually opting for that which was deemed most feasible: *"I always think focusing on reducing emissions is the key bit, but keeping things practically in mind, that could take time to achieve it. So, in the meantime, I think 50% [the scenario with the most CDR] is still an option to go for"* (Milo, Edinburgh G1).

**Region-specific discourses**

When developing appropriate technologies and policy, it is crucial to take local and historical context into account. We identified unique discourses in the Northern Irish, Scottish, and Welsh groups. In Belfast, participants noted the history of government instability and distrust, particularly related to the Renewable Heat Incentive (RHI) scandal, a major political controversy over a failed wood pellet burning scheme which effectively brought down the Northern Irish government in 2020. Another discourse which cut across all three CDR techniques was the importance of farming for the Northern Irish economy and culture: *"It is a big thing here, farming. It has been, historically, a big part of the economy here… And there is a lot of really*

*solid tradition there. When people think of Northern Ireland, you think of industry and steelworks and that kind of thing, but farming has been here a lot longer."* (Ellie & Louise, Belfast G1). Additionally, the Belfast groups discussed peat differently from the other groups, focusing the social equity implications of shifting rural communities away from reliance on peat for jobs and home heating.

In Edinburgh, the most prevalent regional discourse was around land ownership, due to highly unequal patterns of ownership in Scotland, which could have implications for any land-based CDR technique. Lucy (Edinburgh G1) said: *"It's probably owned by landowners, and maybe there are vested interests. I'm not sure how easy it would be to take charge of all that land."* Again, this was connected to equity concerns, with land-based CDR potentially concentrating money and power further into the hands of landowners: *"The use of power is the most dangerous in all three of these… those who already have the means to do so are just going to profit off this and other people are going to suffer day-to-day."* (Meera, Edinburgh G2). In Wales, region-specific discourses were less prevalent, but several participants mentioned the history of Welsh coal, seeing it as a potentially positive thing for biochar *"I think Wales might be more receptive [to biochar] because at one point it was a nation of coal, and people understand coal and it was something that was really ingrained in society."* (Olivia, Cardiff G1). Finally, across all devolved regions, participants discussed the challenges which climate change could pose to biological CDR techniques, particularly in places such as Wales, Northern Ireland and Scotland, which have very high rainfall and regularly experience storms and flooding. Participants' concern about the impacts of climate change thus continued to make itself felt throughout the workshops.

We conducted Kruskal-Wallis H tests (two-tailed) on the survey data to determine whether the four jurisdictions differed in their opinions. We found no statistically significant differences for CDR in general, $\chi^2(3) = 6.514$, $p = 0.089$; biochar, $\chi^2(3) = 7.430$, $p = 0.059$; peatland restoration, $\chi^2(3) = 1.990$, $p = 0.575$; or soil carbon sequestration, $\chi^2(3) = 0.448$,

$p = 0.930$. This is roughly in-line with the workshop ranking tasks, where despite some region-specific discourses we did not see noticeable differences between the jurisdictions in terms of their overall feeling toward CDR or most of the specific techniques. However, Northern Irish survey participants felt significantly more sceptical about BECCS, $\chi^2(3) = 10.60$, $p = 0.014$: this likely reflects the RHI scandal which was mentioned repeatedly in the workshops. For PBC, the survey data (Fig. 2) shows that storing the carbon in the soil would likely be preferred over storing it geologically via BECCS (Mean difference 0.337, Bonferroni post-hoc $p \leq 0.001$ [95% CI = 0.253, 0.420]). This might be particularly the case for Northern Ireland, where the RHI scandal appears to have impacted BECCS perceptions but not SCS.

## Discussion

We find that there is general support for removing carbon from the atmosphere, but that this is tempered with caution over the specific biological techniques we studied.

The majority of our workshop groups and individual participants opted for scenarios with high proportions of carbon removal (CDR) and low proportions of emissions reductions. In the group discussions, the dominant narrative was that this was due to scepticism about emissions reductions, possibly reflecting shifting climate narratives in the UK as some citizens and media outlets become more critical about measures to meet net zero targets[27–29], rather than necessarily positivity about CDR. Emission reductions were seen as largely the domain of individual behaviour change by 'ordinary people', reflecting longstanding diversion of attention away from powerful fossil fuel interests[30,31]. Discussions about emission reductions were greeted with disempowerment, disillusionment, and a general sense of fatigue at the additional pressures of mitigating climate change in a society beset by other crises[32,33]. These results provide important new societal insights into the debate over whether it would be more feasible to focus climate mitigation efforts on demand reduction[34–36]. Although it was not originally the topic of this study, we find that an urgent challenge concerns the shifting of responsibility for climate mitigation away from exhausted individuals who are struggling to make ends meet, and on to polluting industries[33,37,38].

By contrast, CDR was seen as placing more of the burden of responsibility onto powerful actors such as government, industry and the agricultural sector, rather than individuals. In the words of one of our participants: "These are the kinds of things that can make a bigger impact than I ever could". However, there was also some caution about the three specific biological techniques we studied in detail – biochar, perennial biomass crops, and peatland restoration – specifically, whether they would be "worth it". Notions of worth were multi-faceted, extending well beyond monetary cost alone[39,40].

Our methodology enabled us to explore participants' initial affective associations with each of the techniques. In particular, we found that initial high expectations were tempered with multiple concerns, including doubts about whether the techniques would achieve net life-cycle sequestration of carbon (reflecting debates in the CDR literature, e.g. ref. 41). Participants were also concerned about prioritising land-based CDR over other urgent needs like housing (cf. ref. 2), particularly in locations where the capacity for additional public spending was already seen as critically low. CDR techniques may therefore benefit from an 'early leaders' approach, where less vulnerable locations act as early adopters to build knowledge and capacity, which then goes on to benefit locations which initially lack capacity[42].

Communication of scientific uncertainty is one of the biggest challenges in climate policy writ large. Our results, however, suggest that this challenge will be especially pronounced when dealing with CDR techniques which are still being tested, but which will likely need to be rapidly deployed at scale. Questions about the efficacy, durability, co-benefits and trade-offs of CDR techniques are multiplying as the field develops, accompanied by diverse knowledge claims from both scientific and non-scientific actors and organisations[43,44], many of whom have a stake in ongoing scale-up efforts[45]. In such a context, demands for scientific 'certainty' may be strong, but ultimately impossible to fulfil. In line with previous work on climate communication, we found that our stimulus materials appeared to be less trusted when they communicated multiple perspectives on specific CDR techniques[46,47].

That said, our participants also suggested several possible routes forward. For instance, making data from field trial research (including from other countries) more readily accessible, and complementing scientific assessments with more practical forms of knowledge, particularly from farmers and landowners[48]. Perceptions of agency were also crucial, supporting previous work (e.g. refs. 2,49): for example, the idea of community involvement with peatland restoration sparked a sense of agency and empowerment, with one participant reporting that "it makes me want to do conservation volunteering again because it means I can help with climate change, which I didn't even associate before". Another positive and empowered discourse toward the three techniques involved the idea of a portfolio approach, with participants displaying an intuitive understanding of portfolios to maximise benefits and hedge against risks[50,51], and even proposing ideas for how co-deployment could work in specific locations.

From this study's key findings and our participants' discourse, we identify a series of recommendations for developing and deploying socially-robust novel biological CDR techniques. First, there is a need to develop and better communicate data on the things people most want to know about, particularly life-cycle emissions, durability, and land-use requirements. Portfolio approaches where co-benefits are maximised are often well-received. Second, maximising agency (for instance, via community involvement opportunities) will be crucial, and there will be a need to combine scientific and non-scientific sources of expertise to build trust, with traditional and landowner knowledge playing an important role in communicating novel biological CDR to non-experts. Third, when considering deployment locations, it is crucial to consider local contexts and recognise that there is no one-size-fits-all approach; an 'early leader' strategy may help to take pressure off locations with low capacity and strained public resources. Finally, at an overarching level, we advocate regulatory precautions to guard against the potential for large-scale biological CDR deployment to exonerate polluting industries from their responsibilities to mitigate their own emissions[52].

Overall, this work supports a strong role for a broad portfolio of CDR to meet national climate targets, therefore funding and incentives should be ramped up. However, we should not assume that biological CDR will automatically encounter more public support.

## Methods

It is widely acknowledged that deliberation can enable participants to express more nuanced and considered responses and opinion-formation, particular for novel or unfamiliar innovation topics[53,54]. Our research design used 8 day-long deliberations, each lasting around 6 hours and involving 6–8 participants per group to enable meaningful discussions. We held four workshops in four locations (see below), in November and December 2023. Each of the workshops was split in two, with different framings of the discussions and of the information given to participants. We used a mixed methods design to ensure both breadth and depth of understanding, gathering two types of quantitative data in addition to the deliberations: questionnaires and group activities in the workshops ($n = 60$), and a nationally-representative UK survey ($n = 2027$).

### Framing the discussions

Researchers have argued for the 'opening up' of topics of discussion rather than closing them down[13,55], and have made attempts to 'unframe' discussions on CDR[22]. However, we also expected our participants to come to the workshop with low levels of prior knowledge on CDR or the specific techniques, therefore information had to be provided to enable a meaningful discussion. We tackled framing effects in two ways: 1) by following proposals for 'unframing', for instance starting with topics people felt familiar with and providing space for them to frame the issues in their own way, before moving on to less familiar topics and providing more stimulus materials; 2) by splitting the workshop groups in two, each with a different

**Table 1 | Characteristics of the information frames received by the two groups**

|  | Group 1: Techno-economic framing | Group 2: Everyday life framing |
|---|---|---|
| Narrative focus | National and global climate goals | Everyday life in a net zero world |
| Priorities for CDR | Carbon removal potential, permanence and durability | Benefits to the local environment and economy |
| Issues elicited | Global issues and those specific to the administrative regions | Issues specific to the local area and living near to a CDR site |

framing approach, enabling us to gather evidence from multiple sides of a topic. Table 1 shows the framings we used. The following methods section describes the way we implemented these in the workshops.

### Workshop activities

To frame the discussions in the first instance according to those in Table 1, we started by discussing familiar topics in an open manner with participants – first, their associations with their jurisdiction (Group 1) and their local area (Group 2), and next their opinions on 'net zero' emissions. The majority of the UK population are familiar with net zero emissions policy (77.8% in our nationally-representative survey). Participants in Group 1 were given a presentation showing climate change impacts and mitigation targets, presented by the facilitator in a top-down way, followed by group discussion; this was intended to broadly reflect historically dominant discourses on CDR (Supplemental 5a)[23,24]. In Group 2, discussions focused on 'everyday life in a net zero world', and were structured as a group discussion with minimal input from the facilitator in the form of open questions (Supplemental 5b). For both groups, this was followed by a scenarios task to explore trade-offs between CDR and emissions reductions; more on this below.

Next, we introduced three biological CDR techniques in a random order: perennial biomass crops, peatland restoration, and biochar. They were chosen to reflect a balance between 'conventional' (peat restoration) and 'novel' (PBC, biochar) biological CDR (cf. ref. 10); however, none of the techniques is currently used at scale for carbon removal in the UK, reducing potential familiarity bias, and in this sense all three might be considered 'novel'. All three techniques are also land-based, therefore are subject to devolved policy-making to Scotland, Northern Ireland, Wales and England, rather than the UK as a whole. It is worth noting that peat is currently a net source of emissions in most places, and therefore there is debate over its inclusion as a CDR technique, since it will be difficult to shift it from a net source to a net sink[56,57]. That said, techniques such as biochar and other novel CDRs are similarly not proven in their ability to verifiably remove carbon at scale.

We used a novel 'multi-sensory' approach to introduce the specific CDR techniques, deliberately opening up beyond purely cognitive associations. First, we invited participants to pass round objects representing the three techniques and asked them to use their senses of smell, hearing and touch to interact with the objects – a pot of peat soil, a lump of biochar, and a piece of miscanthus grass. As they did this, they were asked to talk about their 'first associations' with the object, eliciting emotions, affective associations, and memories. We then provided short vignette descriptions on pieces of card, and gave everyone a few minutes to individually read through them. The vignettes were designed by the research team, and informed by a series of visits to field trials and labs and ethnographic walking interviews with field researchers (being published separately). Each vignette followed the same structure and length, but we varied the points of emphasis to reflect the framings (see Table 1 & Supplemental 5d). Finally, we showed images, using a google image search to avoid inadvertently exerting too much researcher influence over the images used. This also enabled a broad range of images on the screen, rather than just one. We ran the google search in advance, to ensure that all groups saw the same images (Supplemental 5c). The three techniques were introduced to participants in a random order, followed by discussion of each technique before moving on to the next. In total, the groups discussed the techniques for nearly 2 hours, ending with a discussion of cross-cutting issues and a second scenarios exercise (see below). The workshops ended with a short plenary discussion and a Q&A, not analysed here. Full workshop protocol is shown in Supplemental 4.

### Scenario tasks

The two scenario tasks were designed to encourage participants to consider trade-offs between different options for meeting national climate targets. The groups were informed of their respective climate targets: net zero by 2050 in the UK, England and Northern Ireland, net zero by 2045 in Scotland, and a 95% reduction in net emissions by 2050 in Wales[58,59]. The scenario tasks were held at the beginning, before any CDR stimulus materials had been received, and at the end just before the closing plenary and Q&A.

In the first exercise (Supplemental 5a, b), we gave participants the option of four fictional scenarios for the balance of CDR and emissions reductions: 0% CDR (100% emissions reductions); 10% CDR (90% reductions); 25% CDR (75% reductions); 50% CDR (50% reductions). These were not intended to reflect actual scenarios, shown for the UK in Table 2, which are closer to 17% CDR not including planting trees[58]. The 50% scenario is essentially completely unrealistic. However, none of our participants were expected to be aware of this, and giving a choice of options below 20% CDR would have been less engaging and spark less discussion. Participants were told that these were hypothetical scenarios, and we emphasised that we were looking for their initial responses and 'gut feelings', as a means of stimulating discussion, rather than focusing on the numbers or the technical content of CDR scenarios. In essence, the scenarios were to provoke engagement and comparison across different options, rather than to interrogate actual CDR deployment, and our analysis reflects this. That said, it is important to note that including a 50% scenario may have implied that such a scenario might be more plausible than it actually is.

Removals pathways will differ for the devolved jurisdictions, but we kept the scenarios the same across the four groups to facilitate comparison, using the UK figures due to data availability. We also acknowledged that participants might not feel they had enough information to answer, and therefore asked them to think about their 'gut feeling', with each participant being asked to speak at least once and the ensuing discourse used to understand underlying values and priorities. We recorded a 'show of hands' for their preferred scenario at the end of the discussion. In London G2, we did not manage to do a show of hands, therefore this group is not included in the analysis for this section.

In the second exercise, after the detailed CDR discussions, we distributed a lone-working questionnaire which asked: 1) preferences re. the same four scenario options; 2) how much of a role each of the three biological CDR techniques should play in meeting climate targets (Supplemental 6). After filling out the questionnaires, participants were asked to discuss the scenarios again and to try to reach a group consensus to report back in the plenary. Whilst consensus was often not possible, the process of working towards consensus gave detailed and rich qualitative data on the trade-offs being considered. Next, participants were given a set of 9 sticky dots (for the whole group), and asked to work together to assign the dots to the three biological CDR techniques, according to how much of a role each should play in meeting climate targets. Groups were asked to use all 9 dots. Again, the process of forcing trade-offs and group consensus created rich qualitative data.

### Locations and participants

We tested the full protocol and all stimulus materials with a full-length, full-scale pilot workshop with participants selected using local social media and mailing lists. Following the pilot, we hired a third-party recruitment company to recruit 18 participants per workshop from the general population in the four locations, using topic-blind recruitment to avoid self-selection bias.

**Table 2 | UK central 'balanced pathway' to meet the UK's goal of net zero emissions, published by the UK Committee on Climate Change in 2020**

|  | MtCO2e | % |
|---|---|---|
| 2019 total | 522 | 100 |
| Emissions reductions (all sectors) | 422 | 80.1 |
| LULUCF offsets | 31 | 5.9 |
| Forestry | 12 | 2.3 |
| Peatland restoration | 10 | 1.9 |
| Energy crops | 6 | 1.1 |
| Other land-based removals | 3 | 0.6 |
| BECCS | 54 | 10.3 |
| DACCS | 5 | 1.0 |

Used to illustrate proposed proportions of CDR in meeting UK climate targets

Full participant demographics are shown in Supplemental 1. We used quota sampling to ensure a roughly even mix of gender, age and ethnicity. We also attempted to recruit an even mix of political affiliation, by asking who they would vote for if a general election were held tomorrow, and who they voted for in the previous general election in 2019. However, despite the topic-blind recruitment, it proved extremely challenging to recruit people with right-of-centre political affiliations, and all groups were under-represented in this respect. We believe this may be a persistent bias in deliberative research which warrants more attention in future, although it is worth noting that the centre-left prevalence roughly reflects voting trends in Wales and Scotland. For each workshop, we assigned participants to breakout groups in advance based on achieving as much demographic balance as possible, although this was not always entirely possible in practice due to no-shows (12 in total). Workshops were audio recorded and transcribed by a professional transcription company, and transcripts fully anonymised before analysis. Full informed consent in writing was obtained from all participants prior to the research, as was consent to record their information in line with UK Data Protection law. Ethical approval for this study was granted by University of Oxford, School of Geography and Environment research ethics committee, ethics approval no. SOGE C1A 23 78.

The locations were chosen to reflect the four major policy-making jurisdictions in the UK. In particular, we wished to conduct research in Scotland and Northern Ireland, which tend to be underrepresented in work on perceptions of CDR. Northern Ireland in particular has a longstanding lack of social science research on multiple topics[60]. It is worth noting that the UK/England distinction is complex – in climate policy, there exist separate targets for Scotland, Wales and Northern Ireland, and for the UK overall, but land and environment policy is devolved to England. Therefore we refer to 'jurisdictions' rather than 'nations', and our London groups referred to climate targets in 'the UK' rather than 'England'.

### Survey
The survey was piloted using face-to-face cognitive interviewing ($n = 10$)[61] and two online pilots ($n = 200$). The final survey was then distributed to 3910 people by Qualtrics in June-August 2023. We used quotas to obtain a nationally representative UK sample according to age, gender, ethnicity, and region. After data cleaning for duplicates, bot detection, location data, and attention checks, the total sample was $n = 2027$ (see Supplemental 1 for sample demographics & quotas).

Full survey protocol is shown in Supplemental 2. The survey started with general demographic information, used for quota sampling, followed by questions on climate worry, awareness of net zero targets, and opinion of net zero targets. Next, participants received a short paragraph of information on CDR (see Supplemental 2), followed by questions asking about self-reported knowledge and opinion of 10 specific CDR techniques: Direct Air

Capture with Storage (DACCS), Bioenergy with Carbon Capture and Storage (BECCS), afforestation, soil carbon sequestration, peatland restoration, wood in construction, enhanced rock weathering, biochar, ocean alkalinity enhancement, and blue carbon. Analysis in this paper focuses on four of these, corresponding to the stimulus materials provided in the deliberative workshops: BECCS (representing one form of perennial biomass crops), soil carbon sequestration (representing another form of perennial biomass crops), peatland restoration, and biochar. Techniques were described by what they do, with the technical terms in brackets. The median time taken to complete the entire survey was 13.5 minutes, although the survey contained many sections, most of which are not covered in this paper. The full anonymised dataset is available via the UK Data Service[62] (embargoed until November 2025).

Analysis of the data was carried out using IBM SPSS (version 25). To assess participants' level of support toward CDR and the four techniques, we asked "How do you feel about [technique]?". A one-way repeated-measures ANOVA was run on the responses. We used a repeated-measures test because all participants received the same questions about all CDR techniques in a matrix question, therefore we did not have independence of observations. For the four techniques, participants who had answered that they "did not know enough about it to answer" were removed from the analysis, and treated as missing listwise due to the nature of the repeated-measures design, leaving total $n = 1254$. Mauchly's test of sphericity was violated: $\chi^2(9) = 178.47$, $p \leq 0.001$, therefore a Greenhouse–Geisser correction was used (Huhyn-Feltd correction produced almost exactly the same results). A one-way repeated-measures ANOVA was also used to assess differences between participants' self-reported knowledge. Again, Mauchly's test of sphericity was violated: $\chi^2(9) = 320.04$, $p \leq 0.001$, therefore a Greenhouse–Geisser correction was used.

For the regional comparison, the survey was distributed to samples representing the relative population sizes in the jurisdictions, meaning that the data was unevenly distributed across the groups (Supplemental 1). Therefore a non-parametric Kruskall–Wallis test was used to test for differences in the median scores of how participants felt about CDR, BECCS, soil carbon sequestration, peatland restoration, and biochar (5-point scale, 'very positive' to 'very negative', participants answering "don't know enough" removed and treated as missing listwise). For all five tests, visual assessment of a boxplot showed that the distributions were similar for all groups, therefore the test assessed the differences in the medians.

### Reporting summary
Further information on research design is available in the Nature Portfolio Reporting Summary linked to this article.

### Data availability
All source data for this study is publicly available via the UK Data Service at https://reshare.ukdataservice.ac.uk/857507/ Dataset: https://doi.org/10.5255/UKDA-SN-857507.

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

## Acknowledgements

Funding for all authors was from Natural Environment Research Council (NERC), grant code NE/V013106/1. Additional funding for E.C. came from the Leverhulme Trust, grant no. RC-2015-029. We would like to thank David Moats, Manon Burbidge, Shilpa Sanjeevan, Cath Ibbotson and Alex Black who assisted with the running of the workshops, and researchers at the GGR-D Programme for technical advice.

## Author contributions

E.C., L.W. and R.B. conceived the study. E.C., L.W., J.P. and R.B. designed and conducted the data collection. E.C., L.W. and J.P. analysed the data. E.C. took the lead in writing the manuscript. L.W., J.P. and R.B. discussed the results and contributed to the final manuscript.

## Competing interests

The authors declare no competing interests.
