## [Transparent Peer Review file · Communications Earth & Environment]

Carbon removal support is tempered by concerns over whether biological methods are worth it

Corresponding Author: Dr Emily Cox

Version 0:

Decision Letter:

Dear Dr Cox,

Your manuscript titled "Carbon removal support is tempered by caution over biological methods" has now been seen by 2 reviewers, and we include their comments at the end of this message. They find your work of interest, but some important points are raised. We are interested in the possibility of publishing your study in Communications Earth & Environment, but would like to consider your responses to these concerns and assess a revised manuscript before we make a final decision on publication.

We therefore invite you to revise and resubmit your manuscript, along with a point-by-point response that takes into account the points raised. Please highlight all changes in the manuscript text file. When you revise your manuscript, please ensure that the following editorial thresholds are met:

- Clarify whether participants were adequately informed about the unfeasibility of replacing 50% of emissions reductions with CDR.
- Better justifying the organization of the mixed-methods study results that might raise questions about the rationale for focusing on three specific themes instead of exploring other potential ones.

Please submit your point-by-point responses as a separate file, distinct from your cover letter where you can add responses to the Editors' comments that you do not want to be made available to the reviewers. Word files are preferred. We recommend that any figures, tables or graphs that are included in the response to reviewers are also included in the main article or Supplementary Information.

Please use the following link to submit your revised manuscript, point-by-point response to the referees' comments (which should be in a separate document to any cover letter), a tracked-changes version of the manuscript (as a PDF file) and the completed checklist:

Link Redacted

We hope to receive your revised paper within six weeks; please let us know if you aren't able to submit it within this time so that we can discuss how best to proceed. If we don't hear from you, and the revision process takes significantly longer, we may close your file. In this event, we will still be happy to reconsider your paper at a later date, as long as nothing similar has been accepted for publication at Communications Earth & Environment or published elsewhere in the meantime.

Please do not hesitate to contact us if you have any questions or would like to discuss these revisions further. We look forward to seeing the revised manuscript and thank you for the opportunity to review your work.

Best regards,

Mojtaba Fakhraee, PhD
Editorial Board Member
Communications Earth & Environment
orcid.org/0000-0002-2461-6374

Martina Grecequet, PhD
Senior Editor
Communications Earth & Environment

EDITORIAL POLICIES AND FORMATTING

Editorial Policy: [Policy requirements](https://www.nature.com/documents/nr-editorial-policy-checklist.pdf) (Download the link to your computer as a PDF.)

- Behavioural and social science
- Ecological, evolutionary & environmental sciences
- Life sciences

<https://www.nature.com/documents/nr-reporting-summary.zip>

Furthermore, please align your manuscript with our format requirements, which are summarized on the following checklist: [Communications Earth & Environment formatting checklist](https://www.nature.com/documents/commsj-phys-style-formatting-checklist-article.pdf)

and also in our style and formatting guide [Communications Earth & Environment formatting guide](https://www.nature.com/documents/commsj-phys-style-formatting-guide-accept.pdf) .

***** DATA:** Communications Earth & Environment endorses the principles of the Enabling FAIR data project (<http://www.copdess.org/enabling-fair-data-project/>). We ask authors to make the data that support their conclusions available in permanent, publically accessible data repositories. (Please contact the editor if you are unable to make your data available).

All Communications Earth & Environment manuscripts must include a section titled "Data Availability" at the end of the Methods section or main text (if no Methods). More information on this policy, is available at <http://www.nature.com/authors/policies/data/data-availability-statements-data-citations.pdf>.

If a community resource is unavailable, data can be submitted to generalist repositories such as [figshare](https://figshare.com/) or [Dryad Digital Repository](http://datadryad.org/). Please provide a unique identifier for the data (for example a DOI or a permanent URL) in the data availability statement, if possible. If the repository does not provide identifiers, we encourage authors to supply the search terms that will return the data. For data that have been obtained from publically available sources, please provide a URL and the specific data product name in the data availability statement. Data with a DOI should be further cited in the methods reference section.

REVIEWER COMMENTS:

Reviewer #1 (Remarks to the Author):

This mixed-methods study of perceptions of biological carbon removal adds novelty to the literature due to its methods. The fact that there was a deliberative event that asked participants to consider reductions vs. removals is new, to my knowledge. This research question on public views on the preferred balance of CDR and emissions reductions is an important and policy-relevant question which I have not yet seen examined in the literature.

The approach of using two different workshop frames is also new. The sensory props used in the deliberation are also a new methodological exploration (although one might note that the industrial processes to scale these to climate-significant levels have their own, different sensory profiles – have you been to an ethanol biorefinery? You can smell these from far away). You could imagine that the sensory props of these natural materials might have influenced the deliberations towards a preference for natural CDR.

Overall, this reads as a very well-designed study. The supplementary materials are of high quality and will allow researchers to build on this work. There are two things that could benefit from moderate revisions.

The main question that I will expect will loom for readers is if participants were provided with accurate information about the unfeasibility of replacing 50% of emissions reductions with CDR. In other words, what kind of information were they relying on to judge the scalability of these techniques? Were they able to think critically about it, or not? There is an emergent modeling literature on the sustainability limits of CDR which I'm sure the authors are aware of – suffice it to say there is not enough land, water, energy, and so on to pull off such large amounts of CDR. It's reasonable to ask if positing that as a scenario over-suggested that such a scenario could be possible – like just because these expert researchers put that on the table, it must be a thing that could happen.

The other big-picture issue with the study is that it can be hard to know how to organize results from a mixed-methods study, and I felt that with this paper – the results and discussion sections felt a bit randomly organized. I know there are these three themes that were focused on, whether CDR is “worth it”, how tradeoffs were organized, and regional specific discourses – but that made me wonder why these three themes were picked and if there were not also six other themes that could have emerged from this data. I don't have a strong feeling about how it could be done better, though. If the study was framed in terms of research questions and hypotheses, that would be a clear organizing principle. There is a mention of a “second goal” in line 51 (regarding the tradeoffs) and while the first goal is not explicitly called out, I assume that it was simply to get reactions on these biological CDR methods.

You could imagine an introduction that more clearly states the questions / goals and then a parallel organization for the results section. For example, from the introduction, I thought the paper would be more about the affective and sensory dimensions and then reporting any differences between these techno-economic and everyday life frames, but those weren't even discussed. And I would like to read this as a methods paper, because I think there are important innovations there for science – in general, I would like this community to move from a reporting or descriptive mode to focus more on advances in theory and methods, and the introduction gave me hope in that direction, but then the rest of the paper didn't totally deliver on that promise.

Moreover, I didn't totally understand this discussion of whether CDR was “worth it” (in terms of where it came from, or worth it versus what?) and it seems like this discussion is really about tradeoffs, too, but then the next section is called “Navigating trade-offs”, so basically most of the results section is about tradeoffs.

To sum up: This is a solid paper that could become even more valuable with some minor to moderate revisions, in order to (1) create a clearer organization, (2) improve some figures, and (3) analyze the differences between the workshop framings and impact of the sensory cues, and focus more on the lessons learned for methodology overall (unless the authors are trying to write a separate paper on the methods, which I would discourage because there are just too many papers to review and it is getting ridiculous).

Minor points:

- Lines 18-19 — is it not a stretch to say that nature-based CDR solutions occupy an outsized role in the public imagination given how unfamiliar people are with these?

- Relatedly, I would like to see the answers to the questions of awareness and support for net zero reported somewhere, at least in the SI if not in the main text. This is just important data that can be used as a point of comparison in other research. One of the things that isn't discussed much is that people's assessment of tradeoffs hinges on what they know about mitigation and climate impacts. This makes understanding their baseline awareness valuable for interpreting how they view CDR, and if this study is ever repeated, I would encourage gathering more data on both of these. I was glad to see the Paterson et al (2023) paper on net zero populism cited, and I wonder how those net-zero critical or conspiratorial discourses are impacting this discussion of whether CDR or mitigation are “worth it.” That could be unpacked a bit more.

- This finding at the bottom of p. 8, lines 291+, that CDR was seen as placing more of the burden on responsibility onto political actors rather than individuals, is important and could receive more attention – either moving it more prominently up

or adding something in the introduction that prefaces it.

- The authors didn't discuss the implication of the findings for the mitigation deterrence debates, but someone will probably ask them to, so it could be worth getting out ahead of that.

- One way of reading these results is to conclude that participants just didn't think climate change would be such a big deal. Is that something that was discussed? One could read it as people choosing adaptation by default. I mean, climate change / adaptation as part of the tradeoffs wasn't so present in the paper. I wonder if there could even be a figure that maps out all the different types of tradeoffs discussed (I think more conceptual figures would be useful compared to these graphs).

- Figure 3 was extremely cognitively demanding. I would consider if there is a different way to present the point the authors are trying to make here. I didn't find any of the figures particularly valuable, I regret to say. I think it would even be worth consulting with a professional graphic designer to think about how to do these figures differently and how they can truly reflect the main points the authors want to convey. While I have not seen very many research teams do that, it seems like it would be within the scope of acceptability in terms of paid services that improve the legibility of the manuscript, and I hope the authors have time and budget to consider it. I think there is a lot of rich data here that could benefit from visuals.

- I'm not sure the title gets to the most interesting or important finding here, and the authors may want to keep thinking about it. I mean the fact that it says "caution over biological methods" almost reads as if the participants are worried about biological methods in particular - and if you just read the title, you might assume they had been asked about all the methods and found biological ones the most troublesome, which is not what the authors did in this study. So maybe more workshopping would be good.

Reviewer #2 (Remarks to the Author):

This paper's major claim is that there is a strong desire for carbon removal to help with the UK's national climate targets stemming from skepticism about emissions reductions and changes in behavior related to emissions. That said, this study found there was notable caution amongst its respondents related to biological techniques for carbon removal with a recurrent theme of would be these techniques be 'worth it' explaining much of the collected data.

The findings in this paper are quite novel and important with regard to the discourse around climate targets and biological carbon removal techniques. The mixed methods approach this project employed are very extensive, well-designed, and carefully explained. The methods yielded data that provided validity checks on each data 'bucket' which then allowed for more in-depth analysis and explanatory findings. The use of quotes from respondents was particularly useful for giving nuance and detail to the findings.

The discussion section offers sensible recommendations based on the findings although the final recommendation (regarding climate mitigation shifting) reads a bit personal and may undermine some of the high-quality work that the paper accomplishes by inserting something that may be seen as overly political. The reworking of the final recommendation is suggested.

Overall, this is an excellent paper. It will be a worthwhile contribution in the field of understanding perceptions around carbon removal techniques and environmental decision making.

Communications Earth & Environment is committed to improving transparency in authorship. As part of our efforts in this direction, we are now requesting that all authors identified as 'corresponding author' create and link their Open Researcher and Contributor Identifier (ORCID) with their account on the Manuscript Tracking System prior to acceptance. ORCID helps the scientific community achieve unambiguous attribution of all scholarly contributions. You can create and link your ORCID from the home page of the Manuscript Tracking System by clicking on 'Modify my Springer Nature account' and following the instructions in the link below. Please also inform all co-authors that they can add their ORCIDs to their accounts and that they must do so prior to acceptance.

Version 1:

Decision Letter:

Dear Dr Cox,

Your manuscript titled "Carbon removal support is tempered by caution over some biological methods" has now been seen by our reviewers, whose comments appear below. In light of their advice we are delighted to say that we are happy, in principle, to publish a suitably revised version in Communications Earth & Environment.

We therefore invite you to revise your paper one last time to address the remaining concerns of our reviewers. At the same time we ask that you edit your manuscript to comply with our format requirements and to maximise the accessibility and therefore the impact of your work.

EDITORIAL REQUESTS:

****Please take care to match our formatting and policy requirements. We will check revised manuscript and return manuscripts that do not comply. Such requests will lead to delays. ****

SUBMISSION INFORMATION:

OPEN ACCESS:

Communications Earth & Environment is a fully open access journal. Articles are made freely accessible on publication. For further information about article processing charges, open access funding, and advice and support from Nature Research, please visit <https://www.nature.com/commsenv/open-access>

Link Redacted

Best regards,

Mojtaba Fakhraee, PhD
Editorial Board Member
Communications Earth & Environment

Martina Grecequet, PhD
Senior Editor,
Communications Earth & Environment
@CommsEarth

REVIEWERS' COMMENTS:

Reviewer #1 (Remarks to the Author):

The authors have made minor revisions and have given the reviewer comments a substantive engagement. The research goals read clearly.

My remaining comment is minor, but could be important for the reach of the paper. I have to admit that the addition of "some" in the title made me question if the title is truly the best for the piece. It reads as generic (it did before, too, but I noticed it less).

(1) "Caution" connotes risk to people, but that's not really what people here are worried about, right? (compared to studies on "risk" from geologic storage and CO₂ leakage, say). It seems a missed opportunity not to insert some language about this question of it being "worth it" into the title, since that's really the new thing about the paper.

(2) Plus, the way the title reads, it makes me think the study was comparing what people thought about all the kinds of CDR methods, and they found biological methods lacking. But the study had a focus on these particular techniques of biochar, peat, etc. So the title could even be kind of misleading. I hope there is a chance to workshop it a bit further.

Reviewer #2 (Remarks to the Author):

The changes made to this manuscript help with readability, making the methods section more understandable, and softening some of the more declarative language. The addition of a number of citations makes the manuscript even more well researched and aware of the present state of the field. The new paragraph to start the Results section is worthwhile and adds a sharpness to the rest of the section.

Well done elevating an already excellent manuscript!

Response to Reviewers

Manuscript number: COMMSENV-24-3973-T

Title: Carbon removal support is tempered by caution over some biological methods

We would like to express our sincere thanks to the reviewers for taking the time to read our manuscript, and for helping to make this a better paper. We have substantially revised the manuscript, and have responded to the reviewers' comments individually in the table below. We hope that the MS is now suitable for publication in *Communications Earth and Environment*.

REVIEWER 1		
	Comment	Response
1	The main question that I will expect will loom for readers is if participants were provided with accurate information about the unfeasibility of replacing 50% of emissions reductions with CDR. In other words, what kind of information were they relying on to judge the scalability of these techniques? Were they able to think critically about it, or not? There is an emergent modeling literature on the sustainability limits of CDR which I'm sure the authors are aware of – suffice it to say there is not enough land, water, energy, and so on to pull off such large amounts of CDR. It's reasonable to ask if positing that as a scenario over-suggested that such a scenario could be possible – like just because these expert researchers put that on the table, it must be a thing that could happen.	The participants were told that these were hypothetical scenarios, and we emphasised that we were more interested in their gut feeling responses than in the precise numbers. We do agree, however, that no matter how the session is introduced and framed, including a 50% scenario may have suggested that such a scenario could be possible. We have included a sentence to clarify this in the results section [page 6 of the marked up version, 2nd paragraph], and have also expanded on this in the methods section [page 19 of the marked up version, 2nd paragraph of 'scenarios' sub-section], including alerting the reader to the caveat around framing which the reviewer rightly points to. Of course, what is 'possible' is largely a political and economic question, not a physical one, particularly when talking about one country's emissions, but that is somewhat besides the point – the most important question here is whether or not the participants were able to think critically about this question and able to weigh up the trade-offs involved, which they unquestionably were, as shown by the qualitative analysis we present.
2	The other big-picture issue with the study is that it can be hard to know how to organize results from a mixed-methods study, and I felt that with this paper – the results and discussion sections felt a bit randomly organized. I know there are these three themes that were focused on, whether CDR is "worth it", how tradeoffs were organized, and regional specific discourses – but that made me wonder why these three themes were picked and if there were not also six other themes that could have emerged from this data. I don't have a strong feeling about how it could be done better, though. If the study was framed in terms of research questions and hypotheses, that would be a clear organizing principle. There is a mention of a "second goal" in line 51 (regarding the tradeoffs) and while the first goal is not explicitly called out, I assume that it was simply to get reactions on these biological	This is a great suggestion. We have clarified the structure by: 1. Setting out three goals of the research explicitly in bullet points in the introduction. The three goals reflect the three sections of the results. Edited the mention of the 'second goal' in the intro to improve clarity.2. Re-writing the section headings to better reflect the research questions set out in the bullet points.3. Re-phrased the 'navigating trade-offs' section to focus more explicitly on the CDR vs emissions reductions question, and the scenarios exercise.4. Moved the section on 'trade-offs between the techniques' up to the previous section, to better reflect the structuring of the results section around the RQs. Re-worded this section to focus on comparing the techniques and their relative balance within a portfolio, rather than about trade-offs (which could get confused with the section on the scenario exercise).5. Checked and reworded the 'is it worth it' section (now re-titled) to make it clear that this was simply the main theme which emerged from the thematic

	CDR methods.	analysis, rather than a pre-determined analytic framework. 6. Checked the results section throughout to ensure consistency and clarity.
3	You could imagine an introduction that more clearly states the questions / goals and then a parallel organization for the results section. For example, from the introduction, I thought the paper would be more about the affective and sensory dimensions and then reporting any differences between these techno-economic and everyday life frames, but those weren't even discussed. And I would like to read this as a methods paper, because I think there are important innovations there for science – in general, I would like this community to move from a reporting or descriptive mode to focus more on advances in theory and methods, and the introduction gave me hope in that direction, but then the rest of the paper didn't totally deliver on that promise.	Thank you for this comment, we have now clearly stated the overarching goals of the paper and in a way that parallels the organization of the results section (see above). We agree with the reviewer that references to the novel methodology are distracting from the substantive focus of the analysis. We have now qualified the methodological references in the abstract and introduction. We also always intended to write up the methodological findings separately (currently in progress).
4	Moreover, I didn't totally understand this discussion of whether CDR was "worth it" (in terms of where it came from, or worth it versus what?) and it seems like this discussion is really about tradeoffs, too, but then the next section is called "Navigating trade-offs", so basically most of the results section is about tradeoffs.	Hopefully, the restructuring of the results section (see point 2, above) has helped to fix this issue. We have checked to ensure that the definition of "worth it" is clear in the MS, as defined by participants (page 3 & 4, esp. 1 st paragraph of 'exploring attitudes & discourses' subsection). The 'navigating trade-offs' section now explicitly focuses on the scenario exercises on the balance of CDR and emissions reduction.
5	To sum up: This is a solid paper that could become even more valuable with some minor to moderate revisions, in order to (1) create a clearer organization, (2) improve some figures, and (3) analyze the differences between the workshop framings and impact of the sensory cues, and focus more on the lessons learned for methodology overall.	Thank you for your constructive suggestions – we hope that our responses to points 1-4 above, and point 11 below, have addressed these issues.
6	- Lines 18-19 — is it not a stretch to say that nature-based CDR solutions occupy an outside role in the public imagination given how unfamiliar people are with these?	We have reworded this statement (see tracked changes).
7	- Relatedly, I would like to see the answers to the questions of awareness and support for net zero reported somewhere, at least in the SI if not in the main text. This is just important data that can be used as a point of comparison in other research. One of the things that isn't discussed much is that people's assessment of tradeoffs hinges on what they know about mitigation and climate impacts. This makes understanding their baseline awareness valuable for interpreting how they view CDR, and if this study is ever repeated, I	We have now included graphs / data tables on climate worry, belief in anthropogenic climate change, net zero awareness, and net zero support from the survey, in the new Supplemental 7. We used a graph for climate worry, because the survey options were on a slider from 1 to 10, making the data table slightly more challenging to read. We have also discussed this in the main text and included some data at the start of the results section, since it does give crucial context for the results. We also included a short sentence on responses to climate, net zero and mitigation policies in the deliberative workshops, because this was the topic of some of our opening sections.

	would encourage gathering more data on both of these. I was glad to see the Paterson et al (2023) paper on net zero populism cited, and I wonder how those net-zero critical or conspiratorial discourses are impacting this discussion of whether CDR or mitigation are “worth it.” That could be unpacked a bit more.	We did not wish to go into too much detail on this in the main body of the paper, because there isn’t space to do justice to analysing the climate deliberations in the workshops; and because there is already a lot of existing research on perceptions of climate change and net zero. Regarding populism and conspiratorial discourses – we agree that this is a crucial topic for research, and something which needs more research. Unfortunately we did not collect data on this, and conspiratorial beliefs did not occur in the workshop discussions (we did not include prompts on this, but neither were they raised unprompted). Therefore sadly we do not have enough relevant data to unpack this topic in this paper.
8	This finding at the bottom of p. 8, lines 291+, that CDR was seen as placing more of the burden on responsibility onto political actors rather than individuals, is important and could receive more attention – either moving it more prominently up or adding something in the introduction that prefaces it.	Thank you for this suggestion. We have now referenced this finding in the abstract and connected it to the research questions.
9	- The authors didn’t discuss the implication of the findings for the mitigation deterrence debates, but someone will probably ask them to, so it could be worth getting out ahead of that.	The results section dealing with “trade-offs” now establishes an explicit link between our findings and conventional understandings of CDR as a “mitigation deterrent”. Specifically, we highlight a more nuanced argument emerging from our participants, whereby CDR does not deter mitigation, but rather helps to build support for it.
10	- One way of reading these results is to conclude that participants just didn’t think climate change would be such a big deal. Is that something that was discussed? One could read it as people choosing adaptation by default. I mean, climate change / adaptation as part of the tradeoffs wasn’t so present in the paper. I wonder if there could even be a figure that maps out all the different types of tradeoffs discussed (I think more conceptual figures would be useful compared to these graphs).	We have included a short section on findings relating to climate change concern from the survey and deliberative workshops, at the start of the results section, as well as the new graph and data tables in Supplementary 7. In line with other research, we found that participants were concerned about climate change and definitely felt that it should be addressed. However, when it comes to specific policies (including mitigation vs adaptation) people become more ambivalent. There was plenty of discussion about the difficulties being caused by climate change, including the challenges it poses for CDR; particularly in terms of inclement weather and flooding in the devolved regions of the UK. Unfortunately we are right up against the word limit for this paper, so we couldn’t explore this in too much detail, but we have included a couple of sentences on this at the very top of the results section. Our data suggests that it’s not quite accurate to say that participants didn’t think climate change would be such a big deal, or that they are choosing adaptation by default. We would interpret this more as an issue of choice pressure and fatigue, in the context of some immediate economic and societal challenges (plus possibly a bit of related psychological distance). We discuss this issue of fatigue in the discussion section [page 10 of the marked up version, esp. 2nd paragraph of the Discussion section], and it is covered very thoroughly in Thomas et al 2024.
11	- Figure 3 was extremely cognitively demanding. I would consider if there is a different way to present the point the	Unfortunately, we do not have any money left in the project to consult with a graphic designer, and we don’t

	authors are trying to make here. I didn't find any of the figures particularly valuable, I regret to say. I think it would even be worth consulting with a professional graphic designer to think about how to do these figures differently and how they can truly reflect the main points the authors want to convey. While I have not seen very many research teams do that, it seems like it would be within the scope of acceptability in terms of paid services that improve the legibility of the manuscript, and I hope the authors have time and budget to consider it. I think there is a lot of rich data here that could benefit from visuals.	have the expertise to do professional graphic design ourselves. We have, however, paid attention to how we can improve Figure 3 (now Figure 4) to make it less cognitively demanding. We have changed the layout and the labelling, and inserted a clearer title in the caption. We also changed to a monotone colour scheme, because the focus of the graph isn't supposed to be the individual groups, but we still wanted to include that information for interested readers. In line with the journal style, we now included titles for each graph in addition to the caption below, which hopefully also makes them easier to interpret, and also made the captions clearer by repeating the precise survey question.
--	--	---

REVIEWER 2		
	Comment	Response
1	The discussion section offers sensible recommendations based on the findings although the final recommendation (regarding climate mitigation shifting) reads a bit personal and may undermine some of the high-quality work that the paper accomplishes by inserting something that may be seen as overly political. The reworking of the final recommendation is suggested.	We have reworked the final recommendation to make clear that we are calling for regulatory precautions to guard against the potential for the scale-up of biological CDR to be used to exonerate polluting industries from existing responsibilities to mitigate their own emissions. This is supported by our data, and is also in line with a recent report on "Scaling up carbon dioxide removals" published by the European Scientific Advisory Board on Climate Change, who advocate "separate legal targets for emissions, temporary removals from land and permanent removals from novel methods". We have now cited this report in the paper to support our claim in the penultimate sentence.

Response to Reviewers
Manuscript number: COMMSENV-24-3973-B

We would like to express our sincere thanks to the reviewers for taking the time to read our manuscript, and for helping to make this a better paper; and to the editors for accepting our manuscript to *Communications Earth and Environment*.

We have revised the title of the manuscript, in line with the reviewer's final suggestion for changes – see the main MS, and the response in the Editorial Request Table, submitted separately. No further requests for changes were made by the reviewers.

We have completed all the relevant documentation requested by the editors for this final version – see Editorial Request Table.

Thank you once again for your time and effort in considering this manuscript.

Yours,
Emily Cox, Laurie Waller, James Palmer & Rob Bellamy